# High Step-Up Converter Based on Non-Series Energy Transfer Structure for Renewable Power Applications

**DOI:** 10.3390/mi12060689

**Published:** 2021-06-13

**Authors:** Luis Humberto Diaz-Saldierna, Jesus Leyva-Ramos

**Affiliations:** Instituto Potosino de Investigación Científica y Tecnológica, Camino a la Presa San José No. 2055, San Luis Potosí 78216, Mexico; jleyva@ipicyt.edu.mx

**Keywords:** renewable energy sources, dc-dc power electronic converters, energy efficiency

## Abstract

In this paper, a high step-up boost converter with a non-isolated configuration is proposed. This configuration has a quadratic voltage gain, suitable for processing energy from alternative sources. It consists of two boost converters, including a transfer capacitor connected in a non-series power transfer structure between input and output. High power efficiencies are achieved with this arrangement. Additionally, the converter has a common ground and non-pulsating input current. Design conditions and power efficiency analysis are developed. Bilinear and linear models are derived for control purposes. Experimental verification with a laboratory prototype of 500 W is provided. The proposed configuration and similar quadratic configurations are compared experimentally using the same number of components to demonstrate the power efficiency improvement. The resulting power efficiency of the prototype was above 95% at nominal load.

## 1. Introduction

Nowadays, the even increase in energy consumption and the worldwide concern about environmental issues have led to increase the power generation through renewable sources, like photovoltaic and fuel-cell sources [1,2]. The output voltage of these alternative sources is low and unregulated [3,4]; therefore, an interface with DC-DC power converters is needed to obtain a high and regulated output voltage. [5]. On the other hand, these energy sources can suffer permanent damage if high ripples appear in the demanded current [6,7]. According to the above, DC-DC switching power converters for renewable applications should have high-voltage gains, low-input current ripples and high-power efficiencies [8].

The conventional boost converter has been proposed for renewable applications [9,10]; however, larger duty cycles are required to achieve high voltage gains. The above produces large voltage spikes, diode reverse recovery time problems, and high conduction losses on the active switch due to the intrinsic resistances [11,12].

In the open literature, several configurations are available to further extend the voltage gain without large duty cycles. A boost converter with a quadratic conversion ratio is discussed in [13]. This configuration is based on stackable switching cells with a modular structure that can be extended to boost the output voltage. This converter exhibits a pulsating input current; therefore, a coupled capacitor is needed for renewable energy applications [14]. A quadratic boost converter with two conventional boost converters connected in series (QBC-SC) is proposed in [15]. The power flows in cascade from the input source to the first converter; then, the power flows to the second converter and later to the load.

The quadratic relationship of the conversion ratio provides a high-voltage gain for this configuration. A quadratic boost converter with a single switch (QBC-SS) is disscused in [16]. It is similar to the QBC-SC, the power flows in series through the two converters and load; however, it includes only one switch. A QBC-SS that includes a voltage doubler to achieve an extra voltage gain (QBC-TR) is proposed in [17]. It includes a voltage doubler to achieve an extra voltage gain. The voltage doubler consists of two capacitors, two diodes, and a coupled inductor that provides an additional gain respect to the quadratic conversion ratio. A current-fed boost converter with a quadratic voltage gain (QBC-CF) is proposed in [18]. This converter has two operation modes. Energy from the input source is stored in two inductors; after that, the energy is transfer to two output capacitors. The quadratic gain is obtained by a series arrangement of capacitors, where the output voltage is the sum of both capacitor voltages. The number of diodes and the way to charge both inductors have an impact on the power efficiency. A boost converter with an active switched inductor-capacitor (LC) network is proposed in [19]. The configuration consists of two inductors that are charged in parallel across two active switches, one diode, and one capacitor. This configuration uses a non-common grounding. The energy stored in both inductors is transferred in series to the output capacitor and the load. The structure of this configuration allows an enlargement of the conversion ratio respect to the gain of the quadratic boost converter. On the other hand, a coupling capacitor needs to be connected to the input of the converter because the input current has a pulsating waveform. Two boost configurations based on the quadratic and Cuk converters are shown in [20]. These configurations offer an extra voltage gain by adding a factor increment to the quadratic voltage gain. They are classified into two types, depending on the voltage gain of each configuration. Both converters have non-common grounding. An isolated boost converter that includes forward and flyback configurations is proposed in [21]. This converter is based on a quadratic configuration with an extended voltage gain because of the turns ratio of the transformer; however, the input current has a pulsating waveform.

The quadratic boost converter is a feasible option for processing energy generated by renewable sources, because it has a higher voltage conversion ratio. Quadratic conversion ratios can achieve higher voltage gains with duty cycles less than 0.7. This configuration generates a non-pulsating input current, which is critical for renewable sources as fuel and photovoltaic cells. The circuit of the conventional quadratic boost converter (QBC-SC) is depicted in Figure 1.

A major drawback of the conventional quadratic configuration is the series power transfer between the input source and the load, resulting in a low overall efficiency. The above implies a reprocessing of the energy generated by the source; thus, a slow energy propagation between the input of the converter and the load is obtained. On the other hand, energy reprocessing yields an increment on the power losses due to the parasitics of the converter elements. This issue is described in Figure 2, where the power is transferred throughout the two boost converters of the QBC-SC. As can be seen, the power flow is done in series between the input source and the load, where the capacitor $C_{1}$ acts as the input source of the second converter.

A boost converter with a quadratic voltage gain, common ground, and non-series power transfer for better efficiency is proposed in this work. This configuration consists of two conventional boost converters with a transfer capacitor between both converters. This capacitor allows a parallel energy transfer between both converters and the output. The main advantage is an increase in power efficiency due to the non-series power transfer. Moreover, this configuration can be used for processing energy from alternative sources without the need to add a filter for reducing the input current ripple.

The organization of this work is as follows. A description and operation of the proposed configuration are shown in Section 2. The state-steady conditions and the converter design are developed in Section 3. The power efficiency of the proposed configuration is analyzed in Section 4. Bilinear and linear models of the proposed converter are developed in Section 5. Design criteria of the proposed converter are shown in Section 6. Experimental verification with a 500 W laboratory prototype is provided in Section 7. Final remarks are given in Section 8.

## 2. Converter Operation

The proposed configuration is depicted in Figure 3. The input voltage is represented by *E*, the currents and voltages through $L_{1}$ and $L_{2}$ are denoted by $i_{L_{1}}$, $v_{L_{1}}$, $i_{L_{2}}$, and $v_{L_{2}}$, respectively. The voltages and currents through $C_{p}$, and $C_{0}$ are denoted by $v_{C_{p}}$, $i_{C_{p}}$, $v_{0}$, and $i_{C_{0}}$, respectively. On the other hand, the voltages through $S_{1}$, $D_{S1}$, $S_{2}$ and $D_{S2}$ are denoted by $v_{S1}$, $v_{D_{S1}}$, $v_{S2}$, and $v_{D_{S2}}$. Finally, the current through the load *R* is denoted by $I_{0}$.

The analysis is carried out considering the following points:The study is developed for continuous conduction mode operation (CCM).The passive elements ($L_{1}$, $L_{2}$, $C_{p}$, $C_{0}$), the active switches ($S_{1}$, $S_{2}$), and the passive switches ($D_{S1}$, $D_{S2}$) are considered as an ideal components.The active switches operate under the same duty cycle *D*. Thus, $D_{1}=D_{2}=D$, $f_{s}=1/T_{s}$ is the switching frequency, and $T_{s}$ is the period of the PWM signal that drives the active switches.For steady-state operation, $T_{s}=t_{ON}+t_{OFF}$, where $t_{ON}=DT_{s}$ and $t_{OFF}=\left(1-D\right)T_{s}$.

Due to the above assumptions, the proposed configuration operates in two modes. The power transfer from the input to the output of the proposed configuration is shown in Figure 4. As can be seen, a non-series power transfer between the input source and the load occurs due to the transfer capacitor $C_{p}$.

The two operating modes of the converter are described as follows:

Mode I [$t_{ON}$]: Both active switches ($S_{1}$, $S_{2}$) are turned ON. The diodes $D_{S1}$ and $D_{S2}$ are not conducting; then, the input source *E* delivers energy to the inductor $L_{1}$. The inductor $L_{2}$ is charged through capacitor $C_{p}$ by the energy stored in the capacitor $C_{0}$, which also supplies energy to the load. The circuit that describes this operating mode is exhibited in Figure 5.

Mode II [$t_{OFF}$]: Both active switches ($S_{1}$, $S_{2}$) are turned OFF. For this interval, the output capacitor $C_{0}$ and the load *R* are supplied by the energy stored in inductor $L_{1}$, through capacitor $C_{p}$. The inductor $L_{2}$ delivers energy to the output ($C_{0}$ and *R*) through the diode $D_{S_{2}}$. The circuit that describes this operating mode is exhibited in Figure 6.

The analysis, modeling, and the expressions in steady-state are easy to develop in two operating modes. On the other hand, the capacitor $C_{p}$ acts as a power transfer element between the first converter and the output capacitor $C_{0}$. Due to this arrangement, the cascade power transfer between the first and second converter is avoided; thus, there is an improvement in the converter efficiency.

## 3. Converter Analysis

Figure 7 shows the theoretical waveforms of the proposed configuration. The input converter current $i_{L_{1}}$ has a non-pulsating waveform, and its ripple amplitude depends on the inductance value.

### 3.1. Converter Voltage Gain

Applying the volt-second balance principle to $L_{1}$ and $L_{2}$ results in:(1)$$\begin{array}{cc} EDT_{s}+\left(E+V_{C_{p}}-V_{0}\right)\left(1-D\right)T_{s} & =0\\ \left(-V_{C_{p}}+V_{0}\right)DT_{s}-V_{C_{p}}\left(1-D\right)T_{s} & =0 \end{array}$$

By using the last equations, the expression for $V_{C_{p}}$ can be derived:(2)$$\begin{array}{cc} V_{C_{p}} & =V_{0}-\frac{E}{\left(1-D\right)}=DV_{0} \end{array}$$

From the above expression, the voltage gain *M* is:(3)$$\begin{array}{cc} M & =\frac{V_{0}}{E}=\frac{1}{{\left(1-D\right)}^{2}} \end{array}$$

According to Equation (Equation 3), the voltage gain *M* has a quadratic conversion ratio.

### 3.2. Steady-State Conditions

For calculating the inductances $L_{1}$ and $L_{2}$, the period when the active switches are turned ON is analyzed, see Figure 5. For large enough inductances, the currents rise linearly from their minimum value to their maximum value during $t_{ON}$; then, the voltages in the inductor terminals of $L_{1}$ and the inductor terminals of $L_{2}$ are, respectively:(4)$$\begin{array}{cc} V_{L_{1}} & =L_{1}\frac{\Delta I_{L_{1}}}{t_{ON}}=E\\ V_{L_{2}} & =L_{2}\frac{\Delta I_{L_{2}}}{t_{ON}}=-V_{C_{p}}+V_{0} \end{array}$$

Substituting $t_{ON}=DT_{s}$, $V_{C_{p}}=DV_{0}$, and $V_{0}=E/{\left(1-D\right)}^{2}$ into Equation (Equation 4), the expressions for the inductor current ripples are:(5)$$\begin{array}{cc} \Delta i_{L_{1}} & =\frac{ED}{L_{1}f_{s}}\\ \Delta i_{L_{2}} & =\frac{ED}{\left(1-D\right)L_{2}f_{s}} \end{array}$$

Considering the equality between the input and output power $\left(EI_{L_{1}}=V_{0}^{2}/R\right)$, the average current through $L_{1}$ can be computed. On the other hand, the average current of $L_{2}$ can be computed using the relationship $I_{L_{2}}=\left(1-D\right)I_{L_{1}}$; then, the corresponding expressions for $I_{L_{1}}$ and $I_{L_{2}}$ are:(6)$$\begin{array}{cc} I_{L_{1}} & =\frac{E}{R{\left(1-D\right)}^{4}}\\ I_{L_{2}} & =\frac{E}{R{\left(1-D\right)}^{3}} \end{array}$$

The maximum and minimum current values reached by the first and second inductors can be obtained as follows:(7)$$\begin{array}{c} I_{L_{1_{MAX}}}=I_{L_{1}}+\frac{\Delta I_{L_{1}}}{2},\;I_{L_{1_{MIN}}}=I_{L_{1}}-\frac{\Delta I_{L_{1}}}{2}\\ I_{L_{2_{MAX}}}=I_{L_{2}}+\frac{\Delta I_{L_{2}}}{2},\;I_{L_{2_{MIN}}}=I_{L_{2}}-\frac{\Delta I_{L_{2}}}{2} \end{array}$$

In switching converters, it is essential to find out the critical inductance values for operation in CCM [22]. The critical inductance values of the proposed configuration can be computed using the relationships:(8)$$\begin{array}{cc} 0 & =I_{L_{1}}-\frac{\Delta I_{L_{1}}}{2}\\ 0 & =I_{L_{2}}-\frac{\Delta I_{L_{2}}}{2} \end{array}$$
then, the inductances for the CCM operation are:(9)$$\begin{array}{cc} L_{1} & >\frac{DR{\left(1-D\right)}^{4}}{2f_{s}}\\ L_{2} & >\frac{DR{\left(1-D\right)}^{2}}{2f_{s}} \end{array}$$

The charge variation in a capacitor is defined as $\Delta Q_{C}=C\Delta v_{C}$, where $\Delta Q_{C}$ depicted the area under the current curve when the capacitor stores energy, *C* denotes the capacitance, and $\Delta v_{C}$ is the voltage ripple (see Figure 7). For $t_{ON}$, the capacitor $C_{p}$ is charged by the current $i_{L_{2}}$ ($i_{L_{2}}=i_{C_{p}}$). On the other hand, the capacitor $C_{0}$ is charged by the current of $L_{1}$ at time $t_{OFF}$. In this interval, the current through $C_{0}$ is $i_{C_{0}}=i_{L_{1}}-I_{0}$, that is:(10)$$\begin{array}{cc} \Delta Q_{C_{p}} & =\int_{t_{0}}^{t_{1}}i_{L_{2}}\left(t\right)dt=C_{p}\Delta v_{C_{p}}\\ \Delta Q_{C_{0}} & =\int_{t_{1}}^{t_{2}}\left[i_{L_{1}}\left(t\right)-I_{0}\right]dt=C_{0}\Delta v_{C_{0}} \end{array}$$
where $t_{ON}=t_{1}-t_{0}$ and $t_{OFF}=t_{2}-t_{1}$. Calculating $\Delta Q_{C}$ for $C_{p}$ and $C_{0}$, the following expressions for the capacitor ripples are obtained:(11)$$\begin{array}{cc} \Delta v_{C_{p}} & =\frac{V_{C_{p}}}{R{\left(1-D\right)}^{3}C_{p}f_{s}}\\ \Delta v_{0} & =\frac{V_{0}D\left(2-D\right)}{R\left(1-D\right)C_{0}f_{s}} \end{array}$$

According to Equations (Equation 5) and (Equation 11), the inductor current and capacitor voltage ripples can be chosen for a specific amplitude. Selecting a ripple value is essential, especially in the first inductor current ($\Delta i_{L_{1}}$). The above is due to the requirements of renewable sources as in fuel and photovoltaic cells, which do not allow high ripples on the demanded currents. Large magnitude of current ripple in high frequency (>10 kHz) cause degradation of the catalyst of fuel cell plates. Moreover, fuel-cell currents should not have a high pulsating either negative profile.

### 3.3. Stress on Semiconductor Devices

The current stress values on active and passive switches are computed using the Equations (Equation 5),(Equation 6) and (Equation 7). The current stress on $S_{1}$, $D_{S1}$, $S_{2}$, and $D_{S2}$ are:(12)$$\begin{array}{c} I_{S_{1}}=I_{D_{S1}}=\frac{E[2L_{1}f_{s}+DR{\left(1-D\right)}^{4})]}{2R{\left(1-D\right)}^{4}L_{1}f_{s}}\\ I_{S_{2}}=I_{D_{S2}}=\frac{E[2L_{2}f_{s}+DR{\left(1-D\right)}^{2})]}{2R{\left(1-D\right)}^{3}L_{2}f_{s}} \end{array}$$

Using the Figure 5, the voltage stress values on active and passive switches are computed as:(13)$$\begin{array}{cc} V_{S_{1}} & =-V_{D_{S1}}=V_{0}\left(1-D\right)\\ V_{S_{2}} & =-V_{D_{S2}}=V_{0} \end{array}$$

The Equation (Equation 13) shows that the voltage stress on $S_{1}$ and $D_{S1}$ increases when the duty cycle is reduced. The stress on $S_{2}$ and $D_{S2}$ does not depend on the duty cycle.

### 3.4. Comparison with Other Quadratic Converters

A comparison of the proposed converter with other quadratic configurations described in the Introduction section is now given. The number of components and voltage gains of each converter are shown in Table 1. It can be noticed that, QBC-CS, QBC-SS and the proposed converter consist of only eight components, while QBC-CF includes nine components, and QBC-TR twelve components. Moreover, the voltage gains have the same relationship, with the exception of QBC-TR, which depends on the turns ratio of the transformer $n=N_{2}/N_{1}$.

## 4. Study of the Power Efficiency

Now, the analysis and expressions for computing the power losses of each element for the proposed converter are provided. According to [23], the corresponding circuit including the parasitics of all elements of the configuration is exhibited in Figure 8. The resistances $R_{L_{1}}$, $R_{L_{2}}$, $R_{C_{p}}$, $R_{C_{0}}$, $R_{D1}$, $R_{D2}$, $R_{ON1}$, and $R_{ON2}$ are the equivalent series resistance (ESR) of $L_{1}$, $L_{2}$, $C_{p}$, $C_{0}$, $D_{S_{1}}$, $D_{S_{2}}$, $S_{1}$, and $S_{2}$, respectively. The voltages $V_{F_{DS1}}$ and $V_{F_{DS2}}$ are the forward voltage drops of $D_{S_{1}}$, and $D_{S_{2}}$. The gate voltage for $S_{1}$ and $S_{2}$ is $V_{g}$. In the interval $t_{ON}$, both active switches are conducting. The respective circuit for this interval is described in Figure 9. On the other hand, the respective circuit for the interval $t_{OFF}$ (both active switches are not conducting) is depicted in Figure 10.

Using the volt-second balance principle over $L_{1}$ and $L_{2}$, two expressions for $V_{C_{p}}$ are derived. From these two expressions, it is possible to find the relationship for computing the duty cycle ($D_{loss}$) and the losses of each component of the converter. To precisely quantify the losses associated with the parasitics of the converter, it is necessary to recalculate the duty cycle ($D_{loss}>D$). The resulting equation for $D_{loss}$ has a fourth-order behavior:(14)$$\begin{array}{cc} D_{loss}^{4}+aD_{loss}^{3}+bD_{loss}^{2}+cD_{loss}+d= & 0 \end{array}$$
where:$$\begin{array}{cc} a= & -\frac{4RV_{0}+RV_{F_{DS1}}+4RV_{F_{DS2}}+R_{D2}-R_{ON2}V_{0}}{RV_{0}+RV_{F_{DS2}}}\\ b= & \frac{6RV_{0}+3RV_{F_{DS1}}+6RV_{F_{DS2}}+3R_{D2}V_{0}+R_{L_{2}}V_{0}-2R_{ON2}V_{0}+ER}{RV_{0}+RV_{F_{DS2}}}\\ c= & \frac{\left(R_{ON1}+R_{ON2}-2R_{L_{2}}-4R-R_{D1}-3R_{D2}\right)V_{0}+2ER-3RV_{F_{DS1}}-4RV_{F_{DS2}}}{RV_{0}+RV_{F_{DS2}}}\\ d= & \frac{RV_{0}+RV_{F_{DS1}}+R_{D1}V0+RV_{F_{DS2}}+R_{D2}V_{0}+R_{L_{1}}V_{0}+R_{L_{2}}V_{0}-ER}{RV_{0}+RV_{F_{DS2}}} \end{array}$$
to obtain the duty cycle $D_{loss}$, it is necessary to solve Equation (Equation 14) and choose the adequate root.

The expressions for the power losses of each element are shown in Table 2 (conduction and switching losses are included). The diode junction capacitances of $D_{S1}$ and $D_{S2}$ are $C_{j\_DS1}$ and $C_{j\_DS2}$, respectively. The time intervals are $t_{rr1}=t_{r1}+t_{ON1}$, $t_{rr2}=t_{r2}+t_{ON2}$, $t_{ff1}=t_{f1}+t_{OFF1}$, and $t_{ff2}=t_{f2}+t_{OFF2}$, where $t_{r1}$, $t_{ON1}$, $t_{r2}$, $t_{ON2}$, $t_{f1}$, $t_{OFF1}$, $t_{f2}$, and $t_{OFF2}$ are the rise time, turn ON delay time, fall time, and turn OFF delay time of $S_{1}$, and $S_{2}$, respectively. The input capacitances of $S_{1}$ and $S_{2}$ are $C_{iss1}$ and $C_{iss2}$, respectively. The RMS current values can be obtained as:$$\begin{array}{cc} I_{L1_{RMS}}^{2} & =\frac{V_{0}^{2}}{R^{2}{\left(1-D_{loss}\right)}^{4}},\;\;\;I_{L2_{RMS}}^{2}=\frac{V_{0}^{2}}{R^{2}{\left(1-D_{loss}\right)}^{2}}\\ I_{S1_{RMS}}^{2} & =I_{L1_{RMS}}^{2}D_{loss},\;\;\;I_{D1_{RMS}}^{2}=I_{L1_{RMS}}^{2}\left(1-D_{loss}\right)\\ I_{S2_{RMS}}^{2} & =I_{L2_{RMS}}^{2}D_{loss},\;\;\;I_{D2_{RMS}}^{2}=I_{L2_{RMS}}^{2}\left(1-D_{loss}\right)\\ I_{CP_{RMS}}^{2} & =\frac{V_{0}D_{loss}\left(D_{loss}^{2}-D_{loss}+1\right)}{R^{2}{\left(1-D_{loss}\right)}^{6}}\\ I_{C0_{RMS}}^{2} & =\frac{V_{0}^{2}D_{loss}{\left(2-D_{loss}\right)}^{2}}{R^{2}{\left(1-D_{loss}\right)}^{3}} \end{array}$$

The total power loss is:(15)$$\begin{array}{cc} P_{T_{loss}}= & P_{loss\_L1}+P_{loss\_L2}+P_{loss\_CP}+P_{loss\_C0}+P_{loss\_D_{1}}\\ \; & +P_{loss\_D_{2}}+P_{loss\_S_{1}}+P_{loss\_S_{2}}+P_{loss\_dvr1}+P_{loss\_S_{2}}. \end{array}$$

The expression for computing the power efficiency is:(16)$$\begin{array}{c} \eta =\frac{P_{0}}{P_{0}+P_{T_{loss}}} \end{array}$$
where $P_{0}$ is defined as $P_{0}=V_{0}^{2}/R$.

## 5. Modeling of the Converter

The average modeling is commonly used to describe the behavior of power electronic circuits [24]. The PWM-switch model is a useful technique for describing the electrical behavior of DC-DC power converters.

### Nonlinear Average Model of the Converter

The differential equations for each mode can be derived from Figure 5 and Figure 6, respectively. The differential equations for mode I and mode II are :(17)$$\begin{array}{cc} {\dot{i}}_{L_{1}} & =\frac{1}{L_{1}}\left(E\right)\\ {\dot{i}}_{L_{2}} & =\frac{1}{L_{2}}\left(-v_{C_{p}}+v_{0}\right)\\ {\dot{v}}_{C_{p}} & =\frac{1}{v_{C_{p}}}\left(i_{L_{2}}\right)\\ {\dot{v}}_{C_{0}} & =\frac{1}{C_{0}}\left(-i_{L_{2}}-\frac{v_{0}}{R}\right) \end{array}$$
(18)$$\begin{array}{cc} {\dot{i}}_{L_{1}} & =\frac{1}{L_{1}}\left(v_{C_{p}}-v_{0}+E\right)\\ {\dot{i}}_{L_{2}} & =\frac{1}{L_{2}}\left(-v_{C_{p}}+v_{0}\right)\\ {\dot{v}}_{C_{p}} & =\frac{1}{v_{C_{p}}}\left(i_{L_{2}}\right)\\ {\dot{v}}_{C_{0}} & =\frac{1}{C_{0}}\left(-i_{L_{2}}-\frac{v_{0}}{R}\right) \end{array}$$

By using the switching function as a weighting factor, the average non-linear model can be derived as:(19)$$\left[\begin{array}{c} {\dot{i}}_{L_{1}}\\ {\dot{i}}_{L_{2}}\\ {\dot{v}}_{C_{p}}\\ {\dot{v}}_{C_{0}} \end{array}\right]=\left[\begin{array}{cccc} 0 & 0 & \frac{\left(1-d\right)}{L_{1}} & -\frac{\left(1-d\right)}{L_{1}}\\ 0 & 0 & -\frac{1}{L_{2}} & \frac{d}{L_{2}}\\ -\frac{\left(1-d\right)}{C_{p}} & \frac{1}{C_{p}} & 0 & 0\\ \frac{\left(1-d\right)}{C_{0}} & -\frac{d}{C_{0}} & 0 & -\frac{1}{RC_{0}} \end{array}\right]\left[\begin{array}{c} i_{L_{1}}\\ i_{L_{2}}\\ v_{C_{p}}\\ v_{C_{0}} \end{array}\right]+\left[\begin{array}{c} \frac{1}{L_{1}}\\ 0\\ 0\\ 0 \end{array}\right]e$$

The above description can be generalized as:(20)$$\begin{array}{cc} \dot{x}\left(t\right) & =A(d)x+B(d)e \end{array}$$
where the state vector is $x\left(t\right)={\left[i_{L_{1}},i_{L_{2}},v_{C_{p}},v_{0}\right]}^{⊤}\in R_{+}^{4}$, the control signal d∈(0,1), and the input voltage e∈R. The model described in (Equation 19) is bilinear, since the signal *d* is multiplying to all state variables.

Linealization of non-linear systems is a useful technique for analyzing and controlling complex high-order dynamical systems. This process describes the converter behaviour to small perturbations around an operation point, where the perturbations are applied to the input signals [25].

According to above, each state variable and the input signal are the sum of DC and AC components, which can be decomposed as:(21)$$\begin{array}{cc} i_{L_{1}} & =I_{L_{1}}+{\tilde{i}}_{L_{1}}\\ i_{L_{2}} & =I_{L_{2}}+{\tilde{i}}_{L_{2}}\\ v_{C_{p}} & =V_{C_{p}}+{\tilde{v}}_{C_{1}}\\ v_{C_{0}} & =V_{C_{0}}+{\tilde{v}}_{C_{0}}\\ d & =D+\tilde{d}\\ e & =E+\tilde{e} \end{array}$$
where $I_{L_{1}}$, $I_{L_{2}}$, $V_{C_{p}}$, $V_{C_{0}}$, *D*, *E* represent the DC components, and ${\tilde{i}}_{L_{1}}$, ${\tilde{i}}_{L_{2}}$, ${\tilde{v}}_{C_{p}}$, ${\tilde{v}}_{C_{0}}$, $\tilde{d}$, $\tilde{e}$ are the AC components. In steady state, the AC components are equal to zero.

Linearizing around an equilibrium point
(22)$$\varepsilon :=\left(I_{L_{1}},I_{L_{2}},V_{C_{p}},V_{C_{0}}\right)\in R_{+}^{4}$$

The linear representation of the systems shown in (Equation 19) can be rewritten as:(23)$$\begin{array}{cc} \dot{\tilde{x}} & =A\tilde{x}+B\tilde{u} \end{array}$$
where $\tilde{x}\in R_{+}^{4}$ is the state vector, and $\tilde{u}={\left[\tilde{d},\tilde{e}\right]}^{⊤}\in R_{+}^{2}$ is the vector with inputs. *A* is a constant matrix in $R^{4X4}$, and *B* is a constant matrix in $R^{4X2}$. The average linear time-invariant model is:(24)$$\begin{array}{cc} \left[\begin{array}{c} {\dot{\tilde{i}}}_{L_{1}}\\ {\dot{\tilde{i}}}_{L_{2}}\\ {\dot{\tilde{v}}}_{C_{p}}\\ {\dot{\tilde{v}}}_{C_{0}} \end{array}\right]= & \left[\begin{array}{cccc} 0 & 0 & \frac{\left(1-D\right)}{L_{1}} & -\frac{\left(1-D\right)}{L_{1}}\\ 0 & 0 & -\frac{1}{L_{2}} & \frac{D}{L_{2}}\\ -\frac{\left(1-D\right)}{C_{p}} & \frac{1}{C_{p}} & 0 & 0\\ \frac{\left(1-D\right)}{C_{0}} & -\frac{d}{C_{0}} & 0 & -\frac{1}{RC_{0}} \end{array}\right]\left[\begin{array}{c} {\tilde{i}}_{L_{1}}\\ {\tilde{i}}_{L_{2}}\\ {\tilde{v}}_{C_{p}}\\ {\tilde{v}}_{C_{0}} \end{array}\right]\\ \; & +\left[\begin{array}{cc} \frac{E}{L_{1}\left(1-D\right)} & \frac{1}{L_{1}}\\ \frac{E}{L_{2}{\left(1-D\right)}^{2}} & 0\\ \frac{E}{RC_{p}{\left(1-D\right)}^{4}} & 0\\ -\frac{E(2-D)}{RC_{0}{\left(1-D\right)}^{4}} & 0 \end{array}\right]\left[\begin{array}{c} \tilde{d}\\ \tilde{e} \end{array}\right] \end{array}$$

## 6. Converter Design

Now, the design of a 500 W power converter using the procedure shown in Section 3 is now described. The input voltage *E* is 30 V, the output voltage $V_{0}$ is set to 220 V, and the load *R* is 96.8Ω. A switching frequency of $f_{s}=100$ kHz is selected with an ideal duty cycle of D=0.63. The selection criterion of the switching frequency was made based on the sizing of the passive elements, which increase in value with low switching frequencies. Due to the above, the power efficiency decreases when high inductance and capacitance values are selected. On the other hand, generally the value of the switching frequency used for medium and high power DC-DC converters ranges from 25 kHz to 100 kHz. The corresponding parameters for the designed converter are shown in Table 3.

The theoretical power efficiency at the nominal load of the proposed configuration is 95.4%, which was obtained using (Equation 14)–(Equation 16). The parameters of Table 2 were computed using the datasheet of the semiconductor devices used used to build the prototypes. The ESR values of the two inductors and two capacitors were measured using the meter model LCR-821 from GW Insteak. The theoretical duty cycle, including the power losses, is $D_{loss}=0.635$.

Figure 11 shows a comparison between the ideal gain obtained with expression (Equation 3) and the gain given in expression (Equation 14), where all losses are included. It can be observed that both plots are similar until the duty cycle reaches 0.8; after that, the non-idealities produce a difference as shown in the plot.

## 7. Experimental Verification

Experimental verification with a 500 W laboratory prototype is provided in this section. The prototype was designed according to the parameters given in Table 3, and it is shown in Figure 12. The prototype for QBC-SC and QBC-SS converter is shown in Figure 13. Both converters were built in the same prototype, only one switch was replaced by a diode and vice versa.

The experimental set-up of the converter is described in Figure 14. The relationship of the voltage gain ($V_{0}$/*E*) is 220 V/30 V, and the peak value of the ramp signal is 5 V. Additionally, QBC-SC (Figure 15) and QBC-SS (Figure 16) prototypes were built to compare the experimental power efficiencies of the three configurations.

For a fair comparison, all converters were built using the same components. For the QBC-SS, the active switch $S_{1}$ was replaced by the Schottky diode DSA90C200HB.

The parameters of the QBC-SC and the QBC-SS are shown in Table 4. The parameters were obtained by using the expressions developed in [26]. The theoretical and experimental plots for the comparison of efficiencies are shown in Figure 17. The teoretical plot was obtained using the Equations (Equation 14)–(Equation 16), while the experimental plots of the three converter were obtained for a voltage gain of 220 V/30 V, and varying the load from 484Ω (100 W) to 96.8Ω (500 W), with steps of 100 W. The voltage and current values were obtained using the osciloscopie model MDO3034 and the current probe model TCP303 from Tektronix. As can be seen, the theoretical and experimental plots for the proposed converter are similar, only with small variations. The experimental efficiency of the proposed converter ranges from 97.8% (20% of nominal load) to 95.1% (nominal load). It can be noticed that the proposed configuration offers an improvement in the power efficiency due to the non-series power transfer. One part of the energy processed by the first converter flows directly to the output through the transfer capacitor without being reprocessed by the second converter. A pie chart of breakdown losses for nominal load is shown in Figure 18. It is clear that the biggest losses are in the diodes and inductors, while the switch losses are not significant due to the low ESR value of both switches.

The inductor and output currents of the prototype are exhibited in Figure 19. The average value of $i_{L_{1}}$ is 17.5 A, the average value of $i_{L_{2}}$ is 6.3 A, and for $I_{0}$ is 2.7 A. In Figure 20, the input and capacitors voltages are exhibited The value of *E* is 30 V, the average transfer capacitor voltage $V_{C_{p}}$ is 139 V, while the average output capacitor voltage $V_{0}$ is 220 V.

The voltage waveforms in the active switch $S_{1}$ and the diode $D_{S1}$ are exhibited in Figure 21. The voltage stress on $S_{1}$ and $D_{S1}$ is 98 V. The voltage waveforms in the active switch $S_{2}$ and the diode $D_{S2}$ are exhibited in Figure 22, where the voltage stress on $S_{2}$ and $D_{S2}$ is 222 V.

Voltage and current ripples of the prototype are exhibited in Figure 23. The value of $\Delta v_{0}$ is 2.8 V (1.30%), the value of $\Delta v_{C_{p}}$ is 2.2 V (1.60%), the value of $\Delta i_{L_{2}}$ is 1.7 A (27%), and the value of $\Delta i_{L_{1}}$ is 3 A (17.1%).

## 8. Concluding Remarks

A step-up configuration with a quadratic conversion ratio and a non-series power transfer is proposed in this work. It consists of two conventional boost converters, with two active switches operating with the same duty cycle. This configuration uses a transfer capacitor to avoid reprocessing power between both converters. The proposed configuration can be used for processing energy from renewable generation systems, due to the wide voltage conversion ratio, non-pulsating input current, and higher efficiency compared to other quadratic configurations. The steady-state values and power efficiency analysis are derived for design purposes. A power converter with 220 V output voltage @ 500 W is designed using the procedure given in this work. Experimental validation with a laboratory prototype is exhibited to prove the theoretical results given within. By comparing the experimental power efficiency with the QBC-SC, and the QBC-SS shows that the proposed configuration offers an improvement in the power efficiency. The average linear time-invariant model is derived. This model is a useful tool for control purposes due to the transfer functions of all state variables can be obtained. An appropriate control technique for controlling the proposed converter is the average current-mode control, which consists of two feedback loops (inner and outer). The inner loop uses the inductor current, while the outer loop uses the output voltage for regulation purposes. This configuration can be constructed using a single active switch; however, there is a reduction in the power efficiency.

## Figures and Tables

**Figure 1 micromachines-12-00689-f001:**
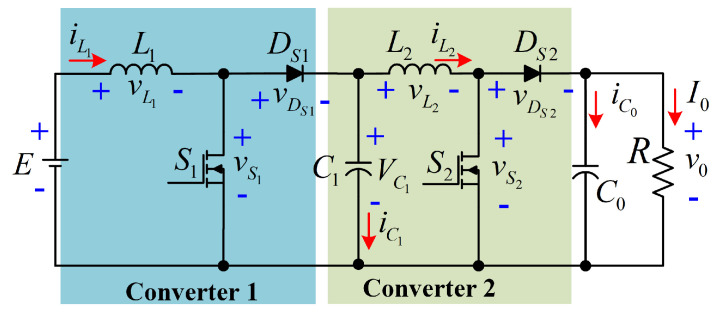
Quadratic boost converter with series power transfer.

**Figure 2 micromachines-12-00689-f002:**
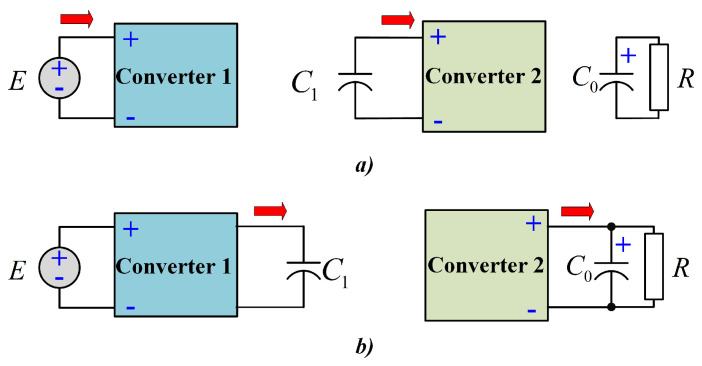
Power flows of the QBC-SC: (**a**) Equivalent scheme when both inductor are storing energy, and (**b**) equivalent scheme when both inductor are delivering energy.

**Figure 3 micromachines-12-00689-f003:**
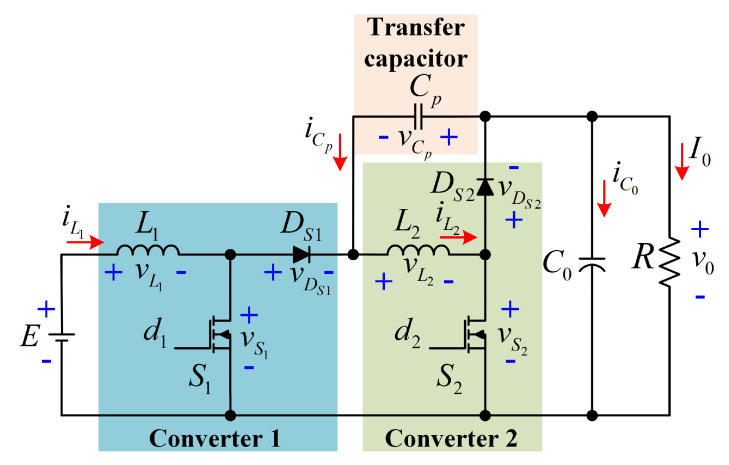
Quadratic boost converter with non-series power transfer.

**Figure 4 micromachines-12-00689-f004:**
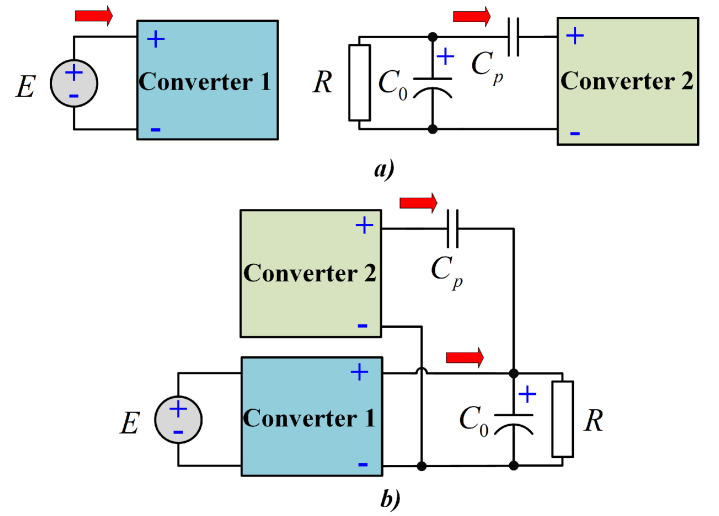
Power flows of the proposed configuration: (**a**) Equivalent scheme when both inductor are storing energy, and (**b**) equivalent scheme when both inductors are delivering energy.

**Figure 5 micromachines-12-00689-f005:**
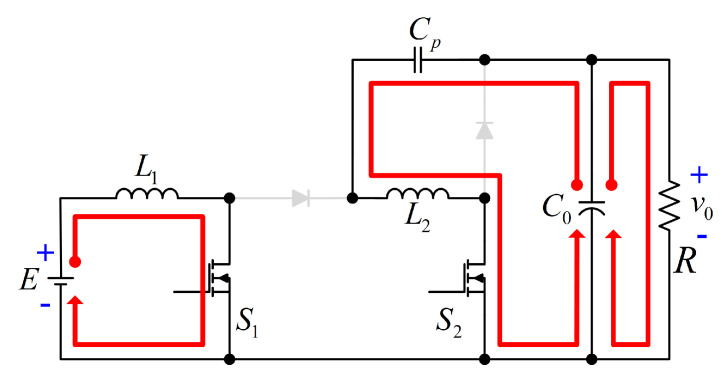
Circuit for the intervale $t_{ON}$.

**Figure 6 micromachines-12-00689-f006:**
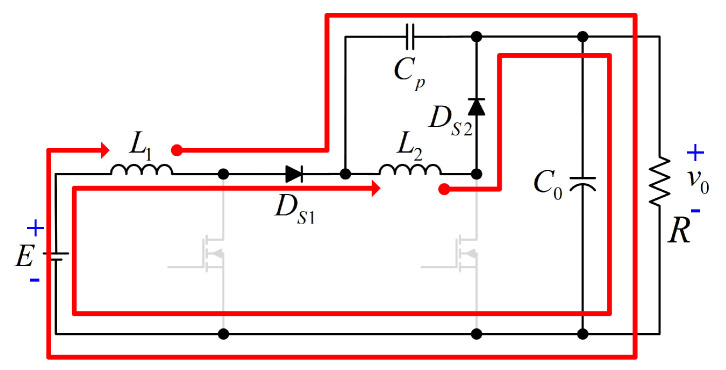
Circuit for the interval $t_{OFF}$.

**Figure 7 micromachines-12-00689-f007:**
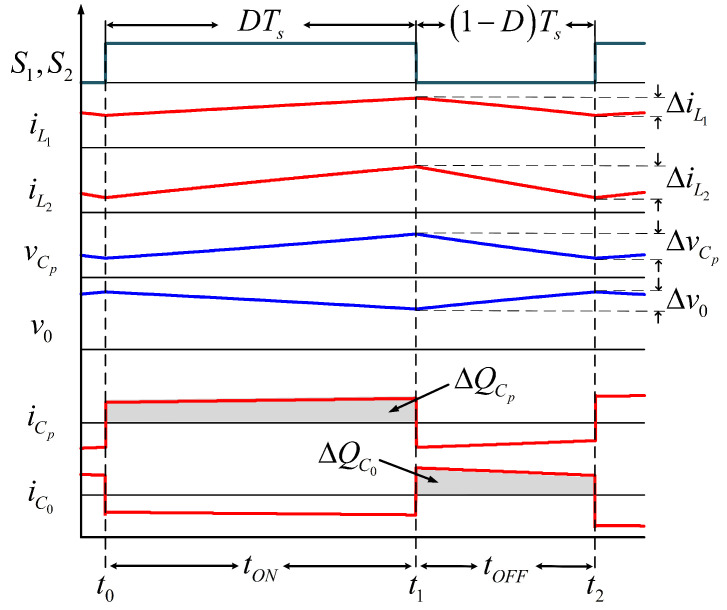
Current and voltage waveforms of the converter in the time $T_{s}$.

**Figure 8 micromachines-12-00689-f008:**
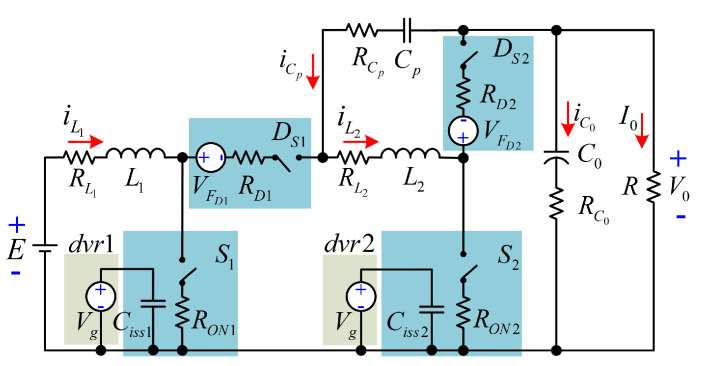
Equivalent circuit with the parasitics of elements.

**Figure 9 micromachines-12-00689-f009:**
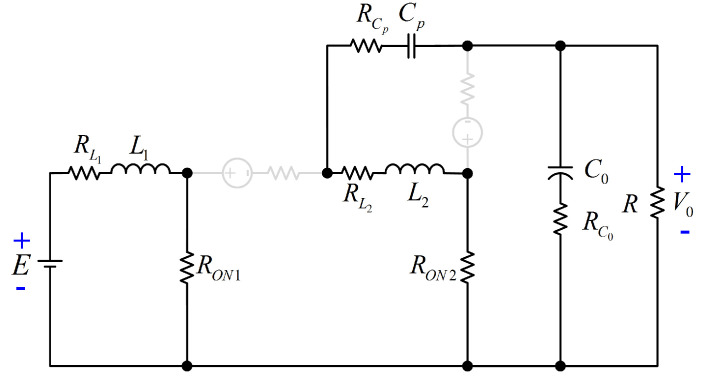
Equivalent circuit for the interval $t_{ON}$.

**Figure 10 micromachines-12-00689-f010:**
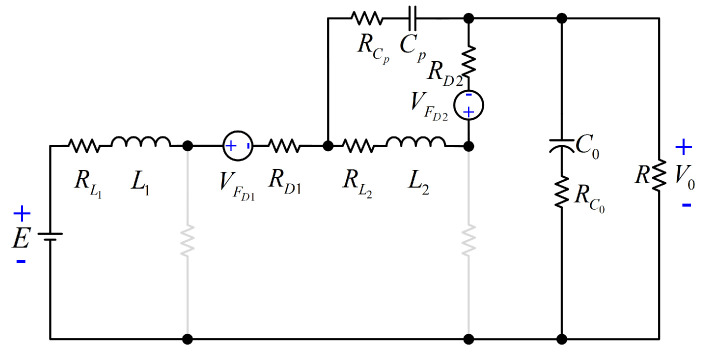
Equivalent circuit for the interval $t_{OFF}$.

**Figure 11 micromachines-12-00689-f011:**
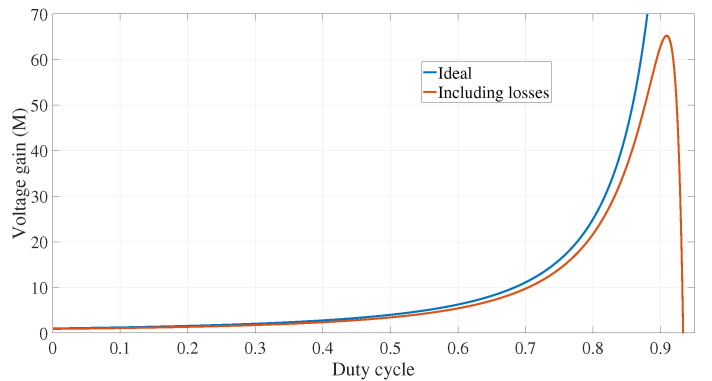
Comparison between the ideal gain and the gain including the power losses.

**Figure 12 micromachines-12-00689-f012:**
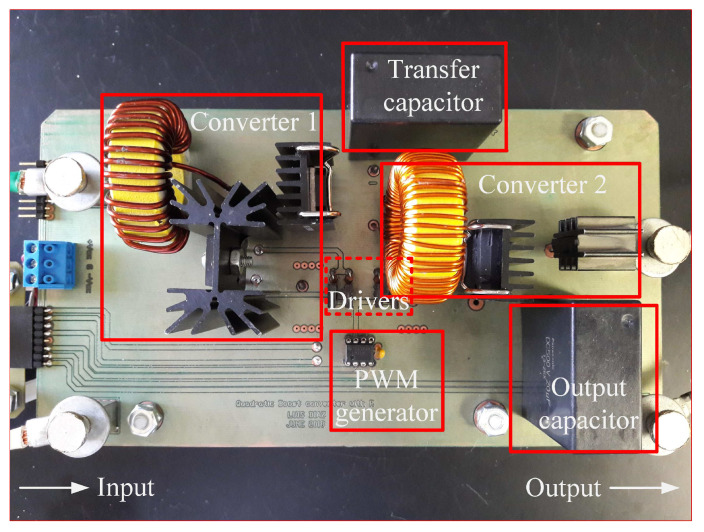
Laboratory prototype of the proposed converter.

**Figure 13 micromachines-12-00689-f013:**
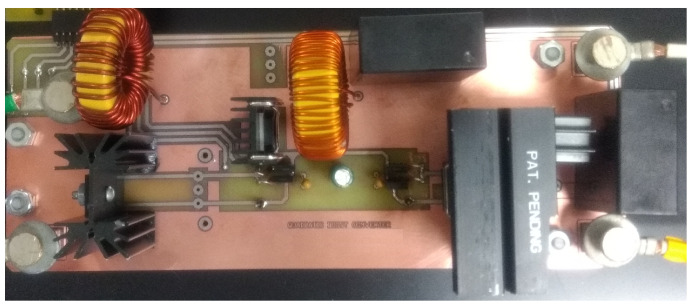
Laboratory prototype of the QBC-SC and QBC-SS converters.

**Figure 14 micromachines-12-00689-f014:**
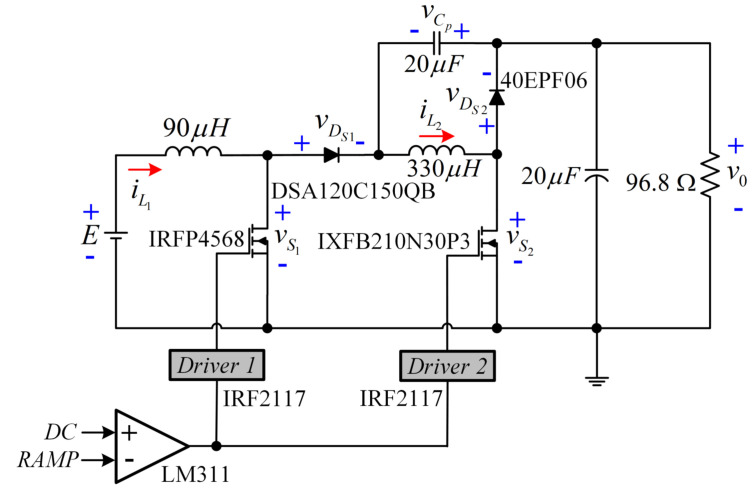
Experimental schematic for the proposed converter.

**Figure 15 micromachines-12-00689-f015:**
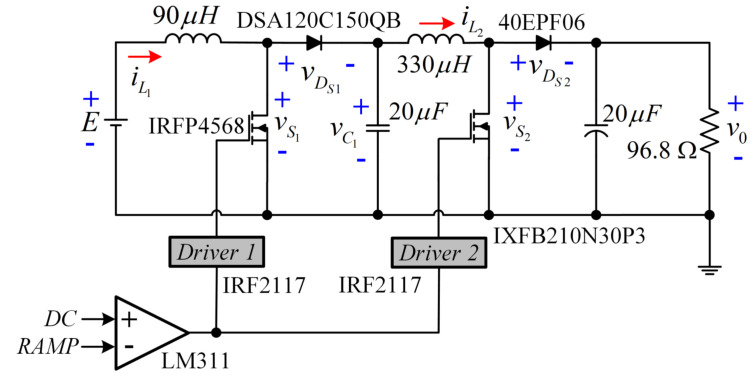
Circuit of the QBC-SC prototype for the experimental efficiency comparison with the proposed converter.

**Figure 16 micromachines-12-00689-f016:**
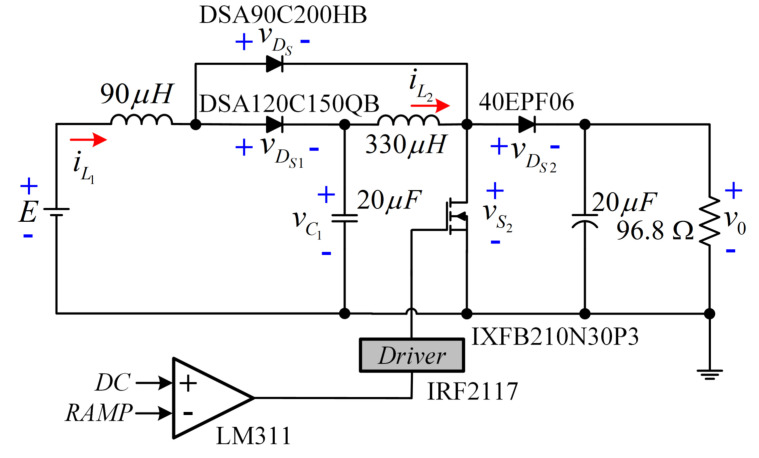
Circuit of the QBC-SS prototype for the experimental efficiency comparison with the proposed converter.

**Figure 17 micromachines-12-00689-f017:**
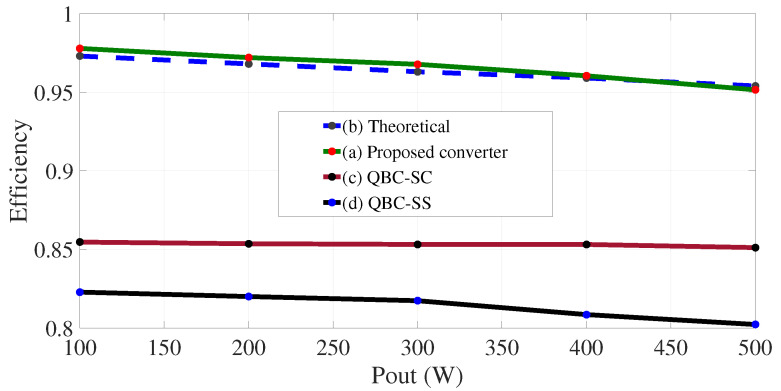
Comparison of power efficiencies. (From top to bottom) (**a**) Proposed converter, (**b**) theoretical efficiency, (**c**) QBC-SC, and (**d**) QBC-SS.

**Figure 18 micromachines-12-00689-f018:**
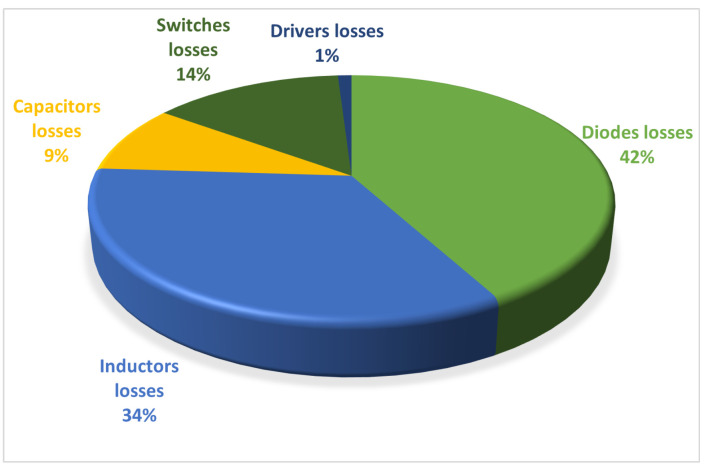
Pie chart of loss breakdown at nominal load.

**Figure 19 micromachines-12-00689-f019:**
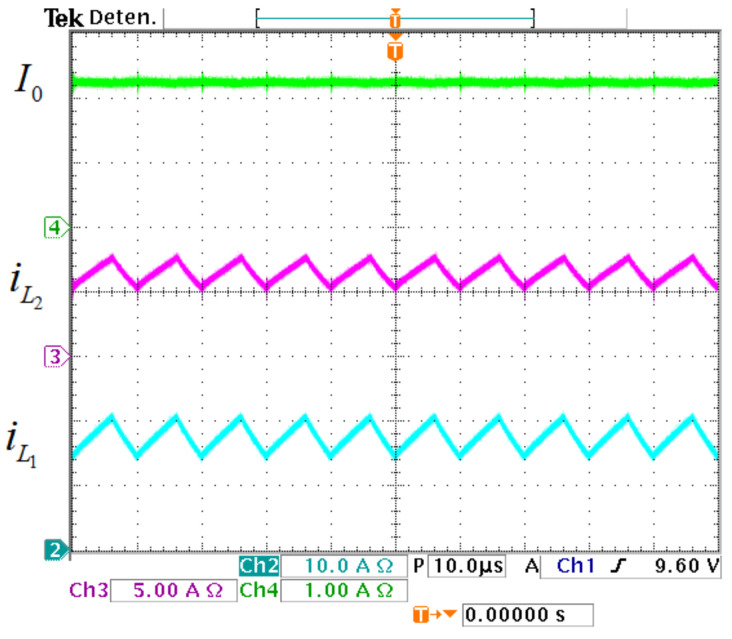
Current waveforms of the prototype. (From top to bottom) Load current $I_{0}$ (1 A/div), second inductor current $i_{L_{2}}$ (5 A/div), and first inductor current $i_{L_{1}}$ (10 A/div) (10 μs/div).

**Figure 20 micromachines-12-00689-f020:**
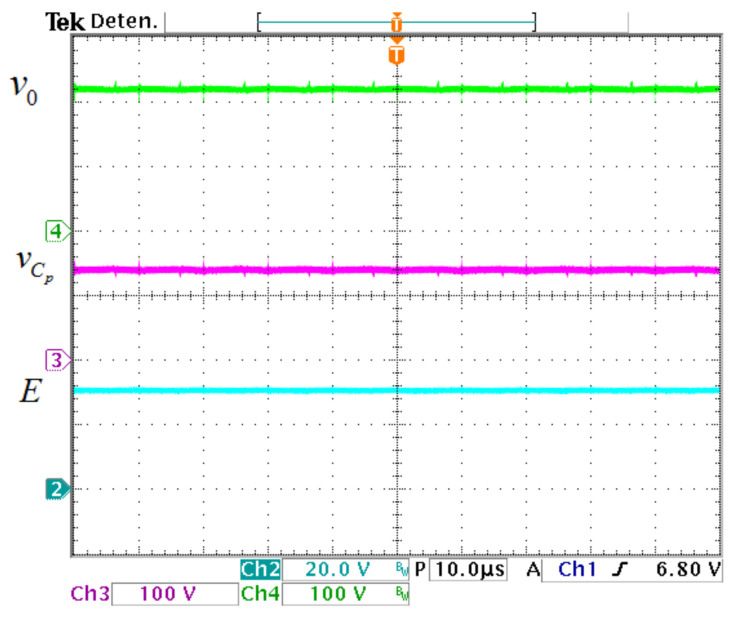
Voltage waveforms of the prototype. (From top to bottom) Output capacitor voltage $v_{0}$ (100 V/div), transfer capacitor voltage $v_{C_{p}}$ (100 V/div), and input voltage *E* (20 V/div) (10 μs/div).

**Figure 21 micromachines-12-00689-f021:**
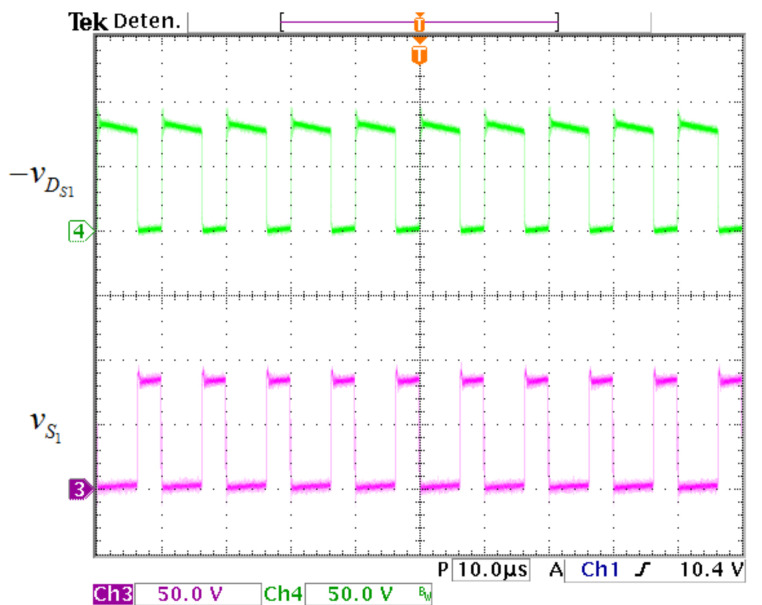
Voltage waveforms in the input of converter. (From top to bottom) Voltage waveforms in $D_{S1}$ (50 V/div) and $S_{1}$ (50 V/div) (10 μs/div).

**Figure 22 micromachines-12-00689-f022:**
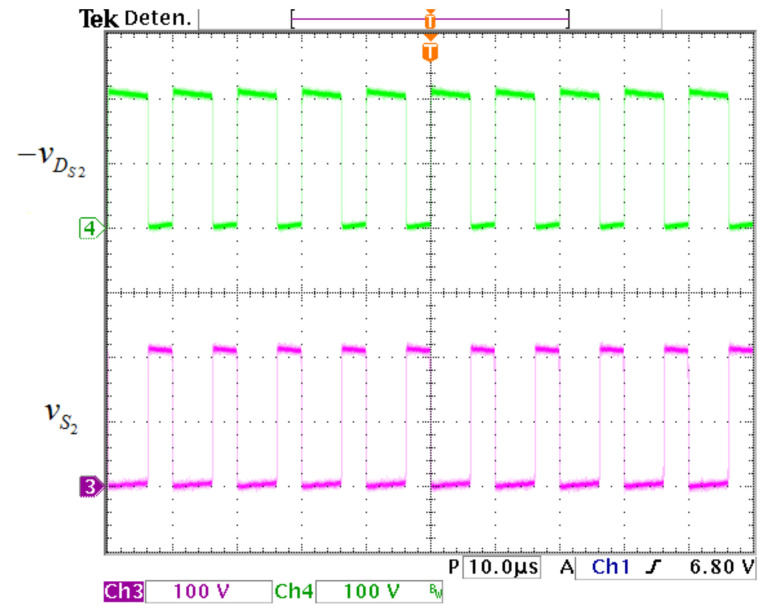
Voltage waveforms in the output of converter. (From top to bottom) Voltage waveforms in $D_{S2}$ (100 V/div) and $S_{2}$ (100 V/div) (10 μs/div).

**Figure 23 micromachines-12-00689-f023:**
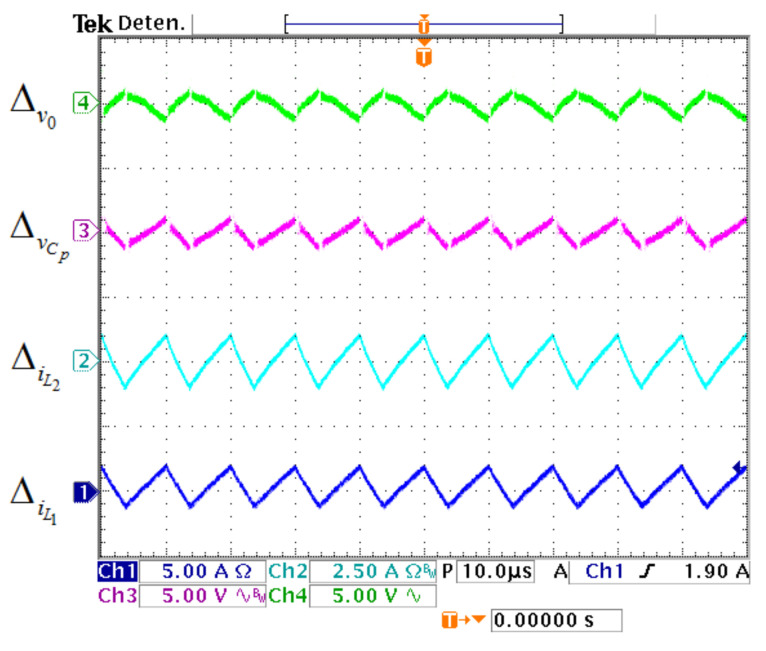
Voltage and current ripples of the prototype. (From top to bottom) Output capacitor ripple $\Delta v_{0}$ (5 V/div), transfer capacitor ripple $\Delta v_{C_{p}}$ (5 V/div), second inductor ripple $\Delta i_{L_{2}}$ (2.5 A/div) and first inductor ripple $\Delta i_{L_{1}}$ (5 A/div) (10 μs/div).

**Table 1 micromachines-12-00689-t001:** Comparison of the proposed converter with other quadratic configurations.

Components	Proposed Converter	QBC-CS Ref. [15]	QBC-SS Ref. [16]	QBC-CF Ref. [17]	QBC-TR Ref. [18]
Capacitors	2	2	2	4	2
Inductors	2	2	2	1	2
Diodes	2	2	3	5	4
Switches	2	2	1	1	1
Transformers	0	0	0	1	0
Gain (CCM)	$\frac{1}{{\left(1-D\right)}^{2}}$	$\frac{1}{{\left(1-D\right)}^{2}}$	$\frac{1}{{\left(1-D\right)}^{2}}$	$\frac{1+n}{{\left(1-D\right)}^{2}}$	$\frac{1}{{\left(1-D\right)}^{2}}$

**Table 2 micromachines-12-00689-t002:** Power losses calculation.

Inductors	$P_{loss\_L1}=I_{L1_{R}MS}^{2}R_{L1}$
	$P_{loss\_L2}=I_{L2_{RMS}}^{2}R_{L2}$
Capacitors	$P_{loss\_CP}=I_{CP_{RMS}}^{2}R_{C_{P}}$
	$P_{loss\_C0}=I_{C0_{RMS}}^{2}R_{C_{0}}$
Diodes	$P_{loss\_D_{1}}=I_{D1_{RMS}}^{2}R_{D_{1}}+I_{L_{1}}V_{F_{D1}}+\frac{1}{2}C_{j\_D1}V_{S_{1}}$
	$P_{loss\_D_{2}}=I_{D2_{RMS}}^{2}R_{D_{2}}+I_{L_{2}}V_{F_{D2}}+\frac{1}{2}C_{j\_D2}V_{S_{2}}$
Switches	$P_{loss\_S_{1}}=I_{S1_{RMS}}^{2}R_{ON1}+\frac{1}{2}\left(t_{rr2}+t_{ff2}\right)I_{L_{2}}V_{S_{2}}$
	$P_{loss\_S_{2}}=I_{S2_{RMS}}^{2}R_{ON2}+\frac{1}{2}\left(t_{rr2}+t_{ff2}\right)I_{L_{2}}V_{S_{2}}$
Drivers	$P_{loss\_dvr1}=\frac{1}{2}C_{iss1}V_{g}^{2}f_{s}$
	$P_{loss\_dvr2}=\frac{1}{2}C_{iss2}V_{g}^{2}f_{s}$

**Table 3 micromachines-12-00689-t003:** Parameters of the proposed converter.

$L_{1}=90\;\mu H$	$L_{2}=330\;\mu H$	$C_{p}=20\;\mu F$	$C_{0}=20\;\mu F$
$I_{L_{1}}=16.6\;A$	$I_{L_{2}}=6.1\;A$	$V_{C_{p}}=138\;V$	$V_{0}=220\;V$
$\Delta i_{L_{1}}=2\;A$	$\Delta i_{L_{2}}=1.5\;A$	$\Delta v_{C_{p}}=2\;V$	$\Delta v_{0}=2.6\;V$
$r_{I}=12.6\%$	$r_{I}=25.2\%$	$r_{V}=1.4\%$	$r_{V}=1.2\%$

where *r_I_* = 100 × (∆*I_L_*/*I_L_*) and *r_V_* = 100 × (∆*V_C_*/*V_C_*).

**Table 4 micromachines-12-00689-t004:** Parameters of the QBC-SC and QBC-SS converters.

$L_{1}=90\;\mu H$	$L_{2}=330\;\mu H$	$C_{1}=20\;\mu F$	$C_{0}=20\;\mu F$
$I_{L_{1}}=16.6\;A$	$I_{L_{2}}=6.1\;A$	$V_{C_{p}}=81\;V$	$V_{0}=220\;V$
$\Delta i_{L_{1}}=2\;A$	$\Delta i_{L_{2}}=1.5\;A$	$\Delta v_{C_{p}}=2\;V$	$\Delta v_{0}=0.7\;V$
$r_{I}=12.6\%$	$r_{I}=25.2\%$	$r_{V}=2.4\%$	$r_{V}=0.3\%$

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
