# Peer review of "High Step-Up Converter Based on Non-Series Energy Transfer Structure for Renewable Power Applications"

_micromachines, 2021, doi:10.3390/mi12060689_

Round 1

Reviewer 1 Report

  1. Lack of sufficient information about the evolution of the proposed circuit: The authors need to clarify how the proposed converter is presented. Maybe add a few Figs before Fig. 1 to show what is the original circuit and what is the contribution.
  2. Lack of sufficient comparison with the existing converters in the literature: There are so many quadratic boost converters in the literature. Please consider adding a comprehensive table and compare them in terms of voltage gain, voltage/current stress on switches/diodes, count of elements, advantages, disadvantages, etc
  3. Lack of controller design for a closed-loop control: The proposed controller is claimed to be presented for renewable energy systems, so there should be state-space analysis along with the controller design and perhaps extra experimental results for a closed-loop controller for renewable energy systems with input source voltage fluctuations.
  4. Lack of sufficient analysis for loss/efficiency calculations: There should be a loss breakdown in form of a pie chart and variations of theoretical efficiency vs experimental efficiency curves. 

Reviewer 2 Report

Comment to the Authors

 The paper presents theoretical explanations and experimental results of a new topology of converter.

As far as I understand you have built three similar converter to be tested and compared.

I have some comments/ issues to clarify.

1.- Influence of the switching frequency

How the switching frequency should be selected?

Has the efficiency of the converter affected by the switching frequency?

Why in your prototypes 100 kHz has been selected?

2.- Prototypes QBS-SC and QBC-SS.

The title of the figures 11 and 12 should be changed to indicated that this are prototypes that have been compared to the proposed converter.

A picture of the other two prototypes should be included in the experimental setup description.

3.- Measurement of the efficiency

In the experimental section of the paper, more details about the measurement of the efficiencies of the three prototypes should be provided.

4.- Ripple of the input current

Has your proposed converted any advantages in the ripple current, in comparison to the other prototypes.

5.- Line 13

“Nowadays, The…” should be changed by “Nowadays, the…”

Round 2

Reviewer 1 Report

The paper has been improved significantly - well done!